# Pancreas Involvement of Extranodal Natural Killer/T-Cell Lymphoma, Nasal Type, Presenting as Acute Pancreatitis: A Case Report

**DOI:** 10.3390/medicina58080991

**Published:** 2022-07-25

**Authors:** Dong Wook Lee, Yun Jeong Kim, Chang Min Cho

**Affiliations:** 1Department of Internal Medicine, School of Medicine, Kyungpook National University, Daegu 41944, Korea; storm5333@naver.com; 2Department of Emergency Medicine, School of Medicine, Kyungpook National University, Daegu 41944, Korea; eumto@naver.com

**Keywords:** extranodal natural killer/T-cell lymphoma, nasal type, acute pancreatitis, fine needle biopsy

## Abstract

Background: The main etiology of acute pancreatitis includes biliary origin and alcohol, although various other causes include drugs (i.e., L-asparaginase) or malignant tumors. Since accurate identification of etiologies is crucial for determining therapeutic planning, the assessment of cause should be performed as early as possible. Case presentation: A 57-year-old Korean man was admitted for chemotherapy. The patient did not drink alcohol for religious reason. 26 days prior to admission, a 4 cm-sized testicular mass was observed in ultrasound and he received right radial orchiectomy. Extranodal natural killer/T-cell lymphoma, nasal type, was diagnosed. After confirming no additional abnormal findings, chemotherapy (using the regimens Dexamethasone, methotrexate, ifosfamide, L-asparaginase, and etoposide) was begun. On Day 8 of chemotherapy, L-asparaginase was started and he complained of sudden onset epigastric pain after 2 days. Acute pancreatitis was diagnosed and, in order to determine if the acute pancreatitis occurred due to L-asparaginase or pancreas involvement of extranodal natural killer/T-cell lymphoma, endoscopic ultrasonography guided fine needle biopsy was performed and observed diffusely infiltrated tumor cells. Therefore, he was given a final diagnosis of acute pancreatitis due to pancreas involvement of extranodal natural killer/T-cell lymphoma, nasal type. Discussion: Acute pancreatitis caused by pancreas involvement of extranodal natural killer/T-cell lymphoma, nasal type, is a very rare disease but can occur during chemotherapy. To identify the cause of acute pancreatitis, endoscopic ultrasonography guided fine needle biopsy can be considered.

## 1. Introduction

Acute pancreatitis is a disease characterized by reversible inflammation of the pancreas, caused by damaged acinar cells [1]. The main etiology of acute pancreatitis includes biliary origin (i.e., gallstones, microlithiasis, or bile sludge) and alcohol, although there are various other causes including anatomic variants, metabolic causes, acute exacerbation of chronic pancreatitis, various drugs (i.e., L-asparaginase), autoimmune disease (i.e., type 1 or 2 autoimmune pancreatitis), trauma, and endoscopic retrograde cholangiopancreatography [2]. Furthermore, in rare cases, benign or malignant pancreatic tumors including cystic neoplasm such as intraductal papillary mucinous neoplasm (IPMN) can be the cause [3]. Since accurate identification of the driving factor among these various etiologies is crucial for determining future therapeutic planning, the assessment of cause should be performed as early as possible after the diagnosis of acute pancreatitis [4]. If anatomical abnormality or a tumorous lesion is observed in radiologic evaluation, or abnormality of blood test in liver function test, autoimmune marker or triglyceride level is investigated, and the etiology can be easily determined. However, if there is no obvious cause, other diagnostic methods should be considered.

In this study, we report a case of acute pancreatitis caused by pancreas involvement of extranodal natural killer/T-cell lymphoma (ENKTL), nasal type, diagnosed by endoscopic ultrasonography (EUS) guided fine needle biopsy (FNB).

## 2. Case Presentation

A 57-year-old Korean man was admitted for chemotherapy. The patient did not smoke or drink alcoholic beverages for religious reasons and exhibited no specific underlying diseases. Under ultrasonography performed due to sudden pain in his right testis, a 4cm-sized testicular mass was observed. Subsequently, he received right radial orchiectomy 26 days prior to hospitalization and was diagnosed with ENKTL nasal type through biopsy from surgical specimen. Paranasal sinus computed tomography (CT) was performed 14 days prior to hospitalization and no abnormal mass or mucosal edema was seen. In addition, CT in neck, chest, and abdomen were performed but no enlarged lymph nodes were observed throughout his entire body. There was no abnormal finding in positron emission tomography (PET) CT (Figure 1A). Bone marrow biopsy results showed normally matured myeloid/erythroid cell and normocellular marrow with reticulin grade 1 and 40% cellularity. At the time of hospitalization, blood test results showed elevated levels of lactate dehydrogenase (306 U/L) and Epstein-Barr virus polymerase chain reaction result (860.40 copies/uL), but all other measurements including complete blood count, liver function test, renal function test, electrolyte, amylase, and lipase levels were within normal range.

The patient did not exhibit disseminated lymphoma throughout his entire body. After confirming no additional abnormal findings, chemotherapy using the same regimens, Dexamethasone, methotrexate, ifosfamide, L-asparaginase, and etoposide (SMILE), was started. On Day 8 of chemotherapy, L-asparaginase (6000 IU/m^2^) was administrated and he complained of sudden onset epigastric pain and tenderness after 2 days. He was hemodynamically stable at the time, and his blood test results were mostly normal with the following values: white blood cell 4400/mm^3^ (neutrophil 67.2%, lymphocyte 23.8%), hemoglobin 12.8 g/dL, platelet 165,000/uL, total bilirubin 0.7 mg/dL, aspartate transaminase 33 IU/L, alanine transaminase 24 IU/L, alkaline phosphatase 63 IU/L, gamma-glutamyl transferase 14 IU/L, triglyceride 123 mg/dL, calcium 8.2 mg/dL, and IgG4 707 mg/L. However, he had elevated levels of amylase (683 U/L) and lipase (1437 U/L), which were three times higher than the normal value. Under abdominal CT (2 days before EUS-FNB), diffuse enlargement of the pancreas was observed compared to the pre-chemotherapy CT (Figure 1B), although neither abnormal mass at pancreas parenchyma nor fluid collection at the peripancreatic area was observed (Figure 2A–C).

After being diagnosed with acute pancreatitis, the patient fasted for therapeutic purposes and underwent EUS to identify the cause. The echogenicity of pancreas parenchyma was mildly decreased, but there was no lesion with different echo levels in the whole pancreas. Main pancreatic duct did not dilate (Figure 3A) and there was no obstructive maneuver on the papillary side. Stones or sludge within gallbladder was not observed (Figure 3B), and common bile duct did not show dilatation (Figure 3C). In order to determine if the acute pancreatitis occurred due to pancreas involvement of ENKTL or L-asparaginase, we performed EUS -FNB using 25-gauge needle (Acquire, Boston Scientific Corp., Natick, MA, USA) (Figure 3D).

We observed microscopically that pancreas acinar cell structure was destroyed by diffusely infiltrated tumor cells (Figure 4A). These tumor cells were medium to large in their size, with irregular pleomorphic hyper-chromatin nuclei (Figure 4B). Tumor cells were strongly positive for cluster of differentiation (CD) 3 (Figure 4C) and CD 56 (Figure 4D) while negative for CD 20 (Figure 4E).

Therefore, he was given a final diagnosis of acute pancreatitis due to diffuse pancreas metastasis of ENKTL, and we resumed chemotherapy using the SMILE regimen. On the second day after fasting, epigastric pain and tenderness started to improve, and the patient resumed oral food consumption. In the follow-up CT performed after 2 months, pancreas enlargement was improved (Figure 5). The patient continued with his chemotherapy with the SMILE regimen, but refused to continue after the third cycle due to severe generalized weakness, and eventually died due to pneumonia. Figure 6 summarized our patient’s hospital course with timeline.

## 3. Discussion

ENKTL is known to arise from the midline fascial structure, such as nasal cavity and nasopharynx [5]. Although it can occur in extra-nasal areas including skin, liver, soft tissue and spleen, ENKTL in pancreas is extremely rare and only one case has been reported until now [6].

Radiological findings of the pancreas involvement of ENTKL are not well known. In the case reported by Liu W et al., this involvement was in the form of a mass in the pancreas head, and the patient received surgical treatment as the mass could not be distinguished from the primary pancreatic cancer [6]. However, the patient in this case exhibited diffuse pancreatic swelling, and we believe that this is caused by acute pancreatitis rather than the pancreas involvement of ENKTL. In addition, since the previous report did not perform EUS, a direct comparison with our case cannot be made.

There are multiple causes of acute pancreatitis, and the identification of cause is crucial for determining the therapeutic plan. Since the patient in this case had no experiences of alcohol and the acute pancreatitis occurred during the hospitalization period, alcohol should not be the cause of acute pancreatitis. Moreover, since the liver function test outcomes were normal under blood test and no stones or sludge were observed in gallbladder or biliary tract under CT or EUS, biliary acute pancreatitis was not the correct diagnosis either. In addition, since all levels of serum triglyceride, calcium level, and IgG4 were in the normal range, the cause could not be metabolic or autoimmune-related factors. Therefore, we supposed that either the administration of L-asparaginase nor the pancreas involvement of ENKTL would be the cause. Although rare, acute pancreatitis with multiple simultaneous causes (such as branch duct type IPMN with type 1 autoimmune pancreatitis and pancreatic intraductal neoplasia) was reported [7], other causes were almost excluded as mentioned above and the confirmation of pancreas involvement of ENKTL was crucial since the continuation of chemotherapy would be dependent on the cause. In other words, if the cause is L-asparaginase, chemotherapy with SMILE regimen should be stopped [8] but, if the pancreas involvement of ENKTL is the cause, completion of chemotherapy cycle should be considered. Therefore, we performed EUS-FNB to identify the cause and determine the patient’s therapeutic plan.

It is known that when performing EUS-FNB, puncturing of pancreas parenchyma without mass increases complications such as pancreatitis [9]. Therefore, to reduce the risk of complications, we used a 25-gauge needle and limited the to-and-fro movement of the needle for up to <15 times. Although we performed the procedure without a rapid on-site pathology review, we minimized the number of puncturing by having an assistant immediately deliver the specimen from FNB to the pathologist and having the specimen reviewed. In this study, two needle passes were sufficient for obtaining an appropriate specimen, and the patient did not experience complications after the procedure.

## 4. Conclusions

Identifying the cause of acute pancreatitis shortly after its diagnosis is crucial since there are various causes that change the therapeutic plan. Although the pancreatic involvement of ENKTL is a very rare disease, it can occur during the chemotherapy with SMILE regimen. More specifically, it can occur in the form of acute pancreatitis, and performing EUS-FNB would be helpful to differentiate from the acute pancreatitis caused by other drugs such as L-asparaginase. However, since EUS-FNB can cause complications, it should be performed with caution and adequate information should be provided to the patient.

## Figures and Tables

**Figure 1 medicina-58-00991-f001:**
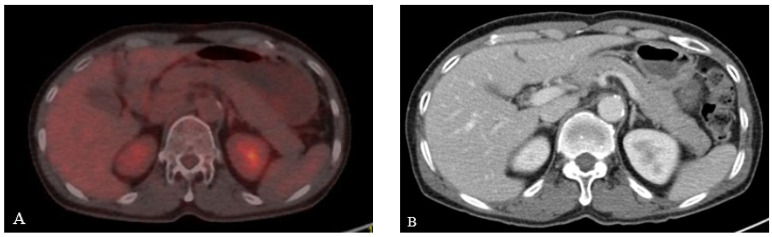
Pancreas findings before chemotherapy. (**A**) There was no finding of presenting malignancy in pancreas on PET CT. (**B**) No abnormal findings suggestive of acute pancreatitis were observed in CT.

**Figure 2 medicina-58-00991-f002:**
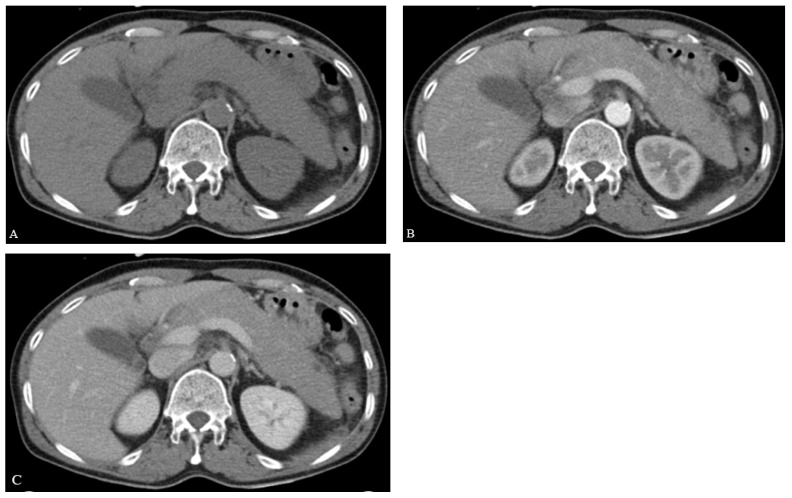
Pancreas finding during chemotherapy. Diffuse pancreas swelling was observed: (**A**) non-enhanced phase (**B**) arterial enhanced phase (**C**) venous enhanced phase.

**Figure 3 medicina-58-00991-f003:**
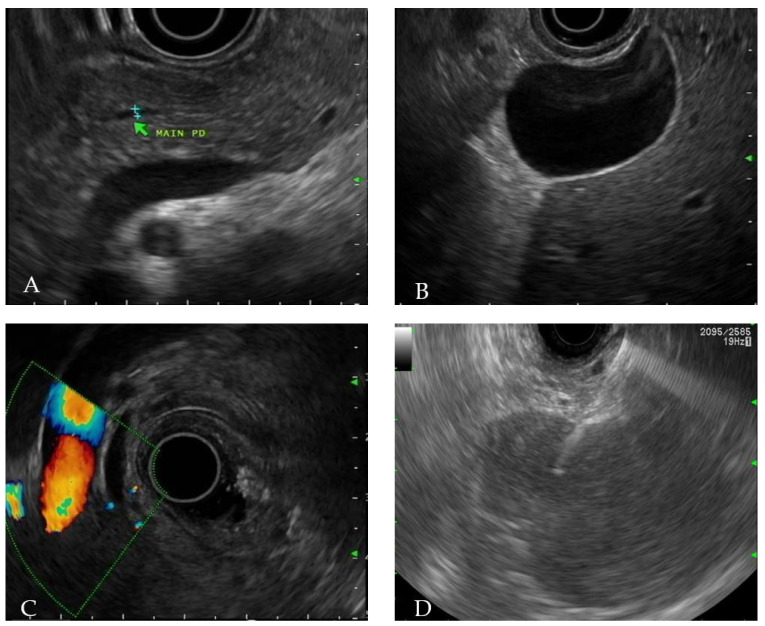
Endoscopic ultrasonography (EUS) findings. (**A**) Main pancreatic duct does not dilate with mild decreased echogenicity of pancreas parenchyma. (**B**,**C**) Observed gallbladder and common bile duct is unremarkable. (**D**) EUS-guided fine needle biopsy is performed at the tail of the pancreas.

**Figure 4 medicina-58-00991-f004:**
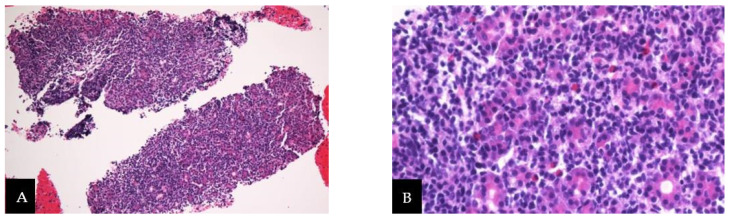
Pathologic findings. (**A**) Acinar cell structure is destroyed by diffusely infiltrated tumor cell (HE, ×100). (**B**) Irregular pleomorphic hyper-chromatin nuclei is observed (HE, ×400). (**C**,**D**) CD 3 and CD 56 is positive in immunohistochemical (IHC) stain. (**E**) CD 20 is negative in IHC stain.

**Figure 5 medicina-58-00991-f005:**
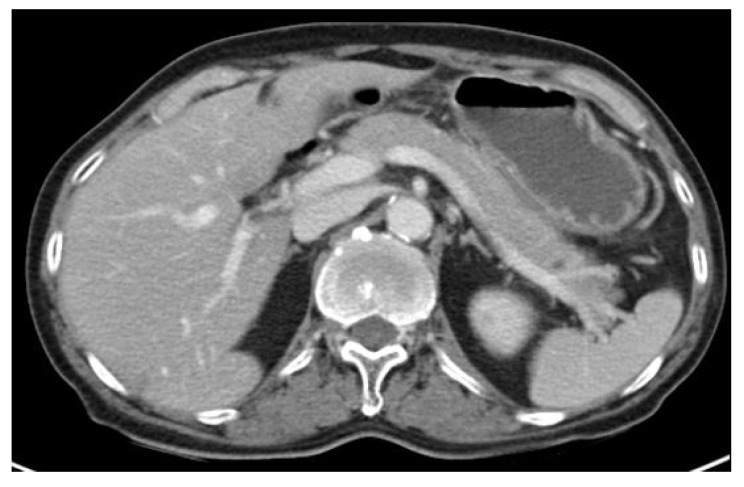
CT finding after 2 months. Pancreatitis is improved and no complication is observed.

**Figure 6 medicina-58-00991-f006:**
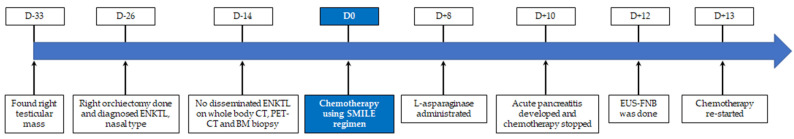
Patient’s hospital course with timeline.

## Data Availability

Not applicable.

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
