# Peer review of "Pancreas Involvement of Extranodal Natural Killer/T-Cell Lymphoma, Nasal Type, Presenting as Acute Pancreatitis: A Case Report"

_medicina, 2022, doi:10.3390/medicina58080991_

Round 1

Reviewer 1 Report

This is a review of ID: medicina-1788906, “Pancreas Involvement of Extranodal Natural Killer/T-cell Lymphoma, Nasal type, Presenting as Acute pancreatitis: A Case Report”.

 The manuscript has been improved according to reviewer's comments, and has great content.

Author Response

This is a review of ID: medicina-1788906, “Pancreas Involvement of Extranodal Natural Killer/T-cell Lymphoma, Nasal type, Presenting as Acute pancreatitis: A Case Report”.

 The manuscript has been improved according to reviewer's comments, and has great content.

--> Thank you very much for your kind reviews.

Reviewer 2 Report

The authors are to be congratulated on an interesting study assessing Pancreas Involvement of Extranodal Natural Killer/T-cell 2 Lymphoma, Nasal type, Presenting as Acute pancreatitis. This is an ongoing field of investigation with regard to abdominal disease. However the script needs improvement prior to publication. The introduction needs more work in terms of explaining why the authors decided to do this study and why they decided to examine those parameters in particular? Obviously the inflammatory response plays a role in this setting but there are many other parameters that they could have chosen? The authors need to be clear to reference previous studies; for exampe, authors could cite the work:          doi: 10.1002/ccr3.2641    to expand conclusion section. 

Author Response

The authors are to be congratulated on an interesting study assessing Pancreas Involvement of Extranodal Natural Killer/T-cell 2 Lymphoma, Nasal type, Presenting as Acute pancreatitis. This is an ongoing field of investigation with regard to abdominal disease. However the script needs improvement prior to publication. The introduction needs more work in terms of explaining why the authors decided to do this study and why they decided to examine those parameters in particular? Obviously the inflammatory response plays a role in this setting but there are many other parameters that they could have chosen? The authors need to be clear to reference previous studies; for exampe, authors could cite the work:          doi: 10.1002/ccr3.2641    to expand conclusion section. 

--> Thank you for your nice and exact comments.

I added the sentence in the introduction (lines 36-41) as follows

"there are various other causes including anatomic variants, metabolic causes, acute exacerbation of chronic pancreatitis, various drugs (i.e L-asparaginase), autoimmune disease (i.e. type 1 or 2 autoimmune pancreatitis), trauma, and endoscopic retrograde cholangiopancreatography [2]. Furthermore, in rare cases, benign or malignant pancreatic tumors including cystic neoplasm such as intraductal papillary mucinous neoplasm (IPMN) and can be the cause"

And I also added the sentence in lines 43-47 as follows.

"If anatomical abnormality or a tumorous lesion is observed in radiologic evaluation or abnormality of blood test in liver function test, autoimmune marker or triglyceride level is investigated, the etiology can be easily determined. However, if there is no obvious cause, other diagnostic methods should be considered."

I added the sentence and cited the reference in the discussion as your recommendation.

"Although rare, acute pancreatitis with multiple simultaneous causes (such as branch duct type IPMN with type 1 autoimmune pancreatitis and pancreatic intraductal neoplasia) was reported [7] but other causes were almost excluded as mentioned above and the confirmation of pancreas involvement of ENKTL was crucial since the continuation of chemotherapy would be dependent on the cause."

All the revised part was expressed in yellow highlight in the manuscript.

Thank you.

Round 2

Reviewer 2 Report

Thaks for modified versione. In my opinion the papaer is acceptable for publication.

This manuscript is a resubmission of an earlier submission. The following is a list of the peer review reports and author responses from that submission.

Round 1

Reviewer 1 Report

COMMENTS TO THE AUTHOR(S)
This is a review of ID: medicina-1719509, “Pancreas Involvement of Extranodal Natural Killer/T-cell Lymphoma, Nasal type, Presenting as Acute pancreatitis: A Case Report”. Though it is interesting, there are some issues to be solved.

Major Comments:

Malignant lymphomas sometimes progress rapidly, and organ invasion is more rapid than in other malignancies. This condition also has a relatively rapid clinical outcome. Although the pancreatic findings were prominent, the possibility of systemic metastasis could not be ruled out. On the other hand, in EUS-FNA performed for acute pancreatitis, puncturing the main pancreatic duct may cause severe injury, and caution should be exercised when puncturing the edematous pancreatic parenchyma. This is an interesting case in which a rapid endoscopic examination allowed an accurate decision about whether to continue treatment for malignant lymphoma.

Minor comments

  • It should be clarified whether the pancreatic metastasis was localized or diffuse. The authors have performed both Convex and Radial EUS. Regarding imaging findings, were there any differences in echo levels between the puncture site and the rest of the pancreatic parenchyma? Was the main pancreatic duct diffusely dilated or was there an obstructive maneuver on the papillary side? Each of these findings is important and should be clearly described.
  • It should be clearly stated whether there were any metastases to other organs.There is a interval between the date of CT imaging and the date of EUS-FNA, whereas malignant lymphoma can easily metastasize to multiple organs in the intervening period. If available, the most recent CT/MRI imaging findings from the EUS-FNA should be noted.
  • Page3, line85 You should note the puncture needle used to perform the EUS-FNB.

Author Response

Thank you for your nice reviews.

The part of revision in manuscript was highlighted in yellow .

Major Comments:

Malignant lymphomas sometimes progress rapidly, and organ invasion is more rapid than in other malignancies. This condition also has a relatively rapid clinical outcome. Although the pancreatic findings were prominent, the possibility of systemic metastasis could not be ruled out. On the other hand, in EUS-FNA performed for acute pancreatitis, puncturing the main pancreatic duct may cause severe injury, and caution should be exercised when puncturing the edematous pancreatic parenchyma. This is an interesting case in which a rapid endoscopic examination allowed an accurate decision about whether to continue treatment for malignant lymphoma.

Minor comments

  • It should be clarified whether the pancreatic metastasis was localized or diffuse --> I revised the sentence in page 4 line 106 "diffuse pancreas metastasis of ENKTL"

  • The authors have performed both Convex and Radial EUS. Regarding imaging findings, were there any differences in echo levels between the puncture site and the rest of the pancreatic parenchyma? --> No, there is no lesion with different echo-levels in whole pancreas. So ". The echogenicity of pancreas parenchyma was mild decreased but there was no lesion with different echo levels in whole pancreas." was added in page 2 line 82.

  • Was the main pancreatic duct diffusely dilated or was there an obstructive maneuver on the papillary side? Each of these findings is important and should be clearly described. --> We already described "Main pancreatic duct did not dilated" in page 2 line 84. And "and there was no obstructive maneuver on papillary side" is added in subsequently.

  • It should be clearly stated whether there were any metastases to other organs.There is a interval between the date of CT imaging and the date of EUS-FNA, whereas malignant lymphoma can easily metastasize to multiple organs in the intervening period. If available, the most recent CT/MRI imaging findings from the EUS-FNA should be noted. --> As you see figure 5, EUS-FNB was performed only 2 days after CT scan and the CT was added in figure 1-B. We added the sentence in page 2 line 74, abdominal CT (2 days before EUS-FNB).

  • Page3, line85 You should note the puncture needle used to perform the EUS-FNB. --> We added the name of needle in page 3 line 88 "using 25-gauge needle (Acquire, Boston Scientific, MA, USA)".

Reviewer 2 Report

The authors presented a case with ENKTL, nasal type, who developed acute pancreatitis during the induction chemotherapy. The pathology confirmed pancreas involvement by the lymphoma. Although pancreas involvement by ENKTL is not that common, some issues are of concern with this current case report. The acute pancreatitis event occurred on Day8, the second day of L-asp administration. Since the pre-treatment images showed no evidence of pancreas, the pancreas involvement is expected to occur or progress DURING the chemotherapy, a sign that strongly indicates the lack of chemotherapy efficacy. The continuation of the same regimen doesn't seem reasonable decision-making. In addition, despite the rarity, pancreas involvement by lymphomas, including ENKTL, has been reported repeatedly. The novelty is not that high. Finally, quite some typos and grammar mistakes are identified; a comprehensive editing process is needed to better the English writing. 

Author Response

Thank you for your review.

The part of revision in manuscript was highlighted in green.

The authors presented a case with ENKTL, nasal type, who developed acute pancreatitis during the induction chemotherapy. The pathology confirmed pancreas involvement by the lymphoma. Although pancreas involvement by ENKTL is not that common, some issues are of concern with this current case report. The acute pancreatitis event occurred on Day8, the second day of L-asp administration. Since the pre-treatment images showed no evidence of pancreas, the pancreas involvement is expected to occur or progress DURING the chemotherapy, a sign that strongly indicates the lack of chemotherapy efficacy. The continuation of the same regimen doesn't seem reasonable decision-making. --> I agree your opinion. However, the patient did not finish the 1st cycle of chemotherapy. So We thought it is not late to make a decision of response for chemotherapy after completion of all chemo-agent administration.

We revised the sentence in page 5 line 140, "In other words, if the cause is L-asparaginase, chemotherapy with SMILE regimen should be stopped [7] but if the pancreas involvement of ENKTL is the cause, completion of chemotherapy cycle should be considered." 

In addition, despite the rarity, pancreas involvement by lymphomas, including ENKTL, has been reported repeatedly. The novelty is not that high. --> That`s right, but it is 1st case of diagnosed by EUS-FNB in diffuse pancreas involvement of ENKTL. 

Finally, quite some typos and grammar mistakes are identified; a comprehensive editing process is needed to better the English writing. --> We revised some expressions.

Round 2

Reviewer 1 Report

The manuscript has been improved according to reviewer's comments, and has great content.